

1       **Mapping land degradation risk due to wind and water erosion**

2       Mahdi Boroughani[1*], Fahimeh Mirchooli[2], Mojtaba Hadavifar[3], Stephanie Fiedler[4]
3               [1]Corresponding letter: Assistant Professor of Research Center for Geoscience
4               and Social Studies, Hakim Sabzevari University, Sabzevar, Iran.
5               Email:m.boroughani@hsu.ac.ir
6               [2]Research Center for Geoscience and Social Studies, Hakim Sabzevari
7               University, Sabzevar, Iran
8               [3] Associate Professor of Environmental Sciences Department, Hakim Sabzevari
9               University, Sabzevar, Iran
10              [4]University of Cologne, Institute of Geophysics and Meteorology, Cologne,
11              Germany

**Abstract**
Land degradation is a cause of many social, economic, and environmental problems.
Therefore identification and monitoring of high-risk areas for land degradation are
necessary. Despite the importance of land degradation, the topic receives often
relatively little attention. The present study aims to create a land degradation map in
terms of soil erosion caused by wind and water erosion of semi-dry land. We focus on
the Lut watershed in Iran encompassing the Lut Desert that is influenced by both
monsoon rainfalls and dust storms. Dust sources are identified using MODIS satellite
images with the help of four different indices to quantify uncertainty. The dust source
maps are assessed with three machine learning algorithms encompassing artificial
neural network (ANN), random forest (RF), and flexible discriminant analysis (FDA)
to map dust sources paired with soil erosion susceptibility due to water. We assess the
accuracy of the maps from the machine learning results with the metric Area Under the
Curve (AUC) of the Receiver Operating Characteristic (ROC). The maps for water and
aeolian soil erosion are used to identify different classes of land degradation risks. The
results show that 43% of the watershed is prone to land degradation in terms of both
aeolian and water erosion. Most regions (45%) have a risk of water erosion and some
regions (7%) a risk of aeolian erosion. Only a small fraction (4%) of the total area of
the region had a low to very low susceptibility for land degradation. The results of this
study underline the risk of land degradation for an inhabited region in Iran. Future work
should focus on land degradation associated with soil erosion from water and storms in
larger regions to evaluate the risks also elsewhere.

Key words: Desertification, Desert-dust sources, Risk susceptibility, Water-induced
soil erosion,




**Introduction**

Land degradation is one of the most pressing environmental issues around the globe.
Several aspects of this issue have been recognized by the United Nations Convention
(Gholami et al. 2019a). Land degradation can be driven by both water and wind, of
which the former can have a stronger impact on soil erosion in short time (Gia et al.
2018). Spatial mapping of risks of land degradation can provide a basis to support
managers and policymakers in risk mitigation and adaptation to aeolian and water
erosion.
Land degradation driven by aeolian erosion is a known problem (Shi et al. 2004). Dust
storms, which are a natural hazard, are associated with soil erosion. This phenomenon
has detrimental impacts in the Earth system, e.g., for food security (Boroughani et al.
2022), human health (Moridnejad et al., 2015), geochemical conditions (Gholami et al.,
2020b), and the Earth's carbon cycle (Gherboudj et al., 2017). Identifying dust sources
as potential areas of dust emission is therefore necessary in developing a better
understanding of land degradation. Spatial mapping of dust source susceptibility areas
(DSSAs) is a crucial step for erosion mitigation and watershed management. Different
approaches for identifying DSSAs exist, e.g., using meteorological data (Yang et al.
2019), numerical modeling (Péré et al. 2018), and remote sensing (Jafari et al. 2021).
Remote sensing can provide worldwide information on aerosol properties (Park et al.
2014). The present study uses MODIS satellite images to detect dust aerosols over the
Lut Desert.
In addition to soil erosion by wind, water-driven soil erosion is a known mechanism for
soil degradation. This kind of soil erosion is a known environmental threat and can
influence both terrestrial and aquatic systems (Halecki et al. 2018, Sun et al. 2014).
Therefore, knowing the spatial distribution of water-induced soil erosion susceptibility
areas (SESA) is also necessary. Several numerical models exist for predictions and risk
evaluations of water-induced soil erosion (Chicas et al., 2016, Gao et al., 2017, Anache
et al., 2018, Gia et al., 2018, Halecki et al., 2018), but none used machine learning to
combine different observational data sets for assessing soil erosion. Machine learning
has emerged as a subfield of data science and helps to better understand environmental
problems (Gholami et al. 2019b). It can integrate data from different sources to create
forecasts and discover patterns (Gholami et al. 2020a). In environmental sciences,



algorithms such as support vector machine, random forest (RF), artificial neural
networks (ANN), and multivariate adaptive regression spline have been applied, e.g.,
for groundwater (Lee et al. 2017), gully erosion (Zabihi et al. 2018), sediment
contamination (Mirchooli et al. 2019), dust sources (Boroughani et al. 2020), landslides
(Youssef and Pourghasemi 2021), floods (Tehrany et al. 2014), and trace elements
(Derakhshan-Babaei et al. 2022).
The aims of the current study are (1) to assess the spatially resolved contribution of soil
erosion by water and wind, and (2) to combine the findings into spatially resolved
information on risks for land degradation.

**2. Data and methods**
The focus of this study is on the Lut watershed situated in the east and southeast of Iran
covering an area of 206242 km$^2$ ( 28º 10' to 32º 30' N latitude and  55º 45' to 61º 15' E
longitude) and is marked in Fig. 1. The region includes the largest desert of the country,
the Lut Desert. The region contributes to the increasing dust concentration of southwest
Asia (Ebrahimi-khusfi et al. 2021). This area is chosen to develop and test the methods
based on regional data on erosion observations with examples shown in Fig. 1a-d. The
methods tested in this article could be later transferred to similar assessments in other
regions. Choosing this region in Iran is further motivated by the impact of land
degradation on the country's economy. It is estimated that land and water degradation
cost Iran about US \$12.8 billion per year which is four percent of the total Gross
Domestic Product (GDP) (Emadodin et al. 2012). It underlines the impacts of land
degradation that goes well beyond impacts on the natural environment.

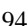
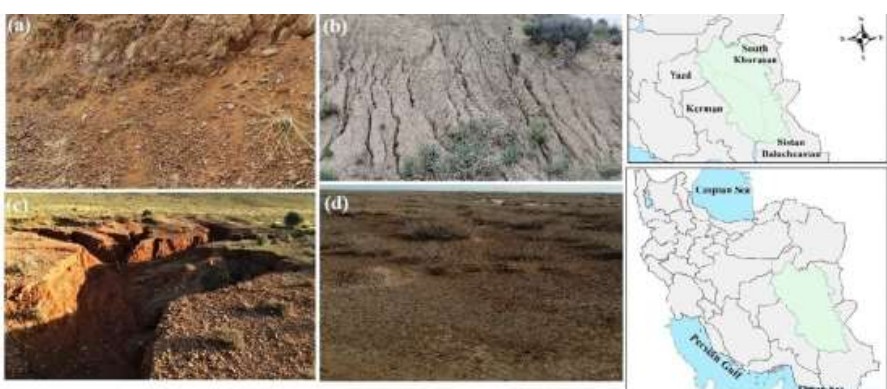




Fig.1 Geographical location of the study watershed. Green shading marks the Lut watershed. The Lut Desert is located in the centre of the watershed. Settlements are primarily situated in the northern and south-western parts. Example of soil erosion in the watershed are sheet erosion (a), rill erosion (b), gully erosion (c), and wind erosion (d).

## 2. 1. Land degradation mapping

Our land degradation zonation consists of three main processing steps, graphically depicted in Fig. 2. At first, spatial mapping of water erosion is conducted (section 2.1.1). In the second step, spatial mapping of dust source susceptibility is carried out with machine learning methods (section 2.1.2). In the last step, the patterns of water erosion and dust source susceptibility are combined to identify risk areas of land degradation (section 2.2.3).

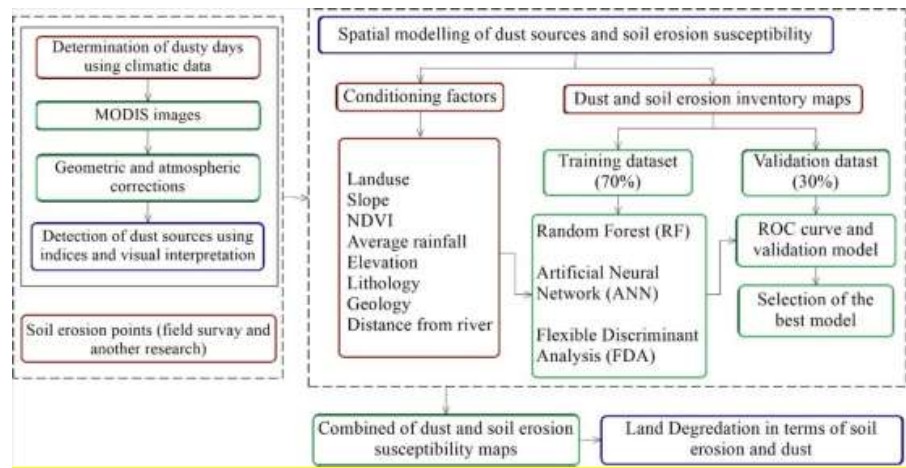

Fig.2 Flowchart of inputs (red boxes), data processing (green boxes), and outputs (blue boxes) in the present study

## 2.1.1 water erosion map

Quantifying the erosion susceptibility of an area requires to determine a spatial distribution of observed water-induced soil erosion that can have different characteristics, e.g., gully erosion, rill erosion, and surface erosion. That information are extracted from data collected during an own field surveys paired with a previous research (Shit et al. 2020). The aim of the field survey for the present study was to identify regions where sheet, rill, and gully erosion took place. This field survey was





carried out in accessible parts of the watershed in April 2020. The data set contains the
type of water-induced soil erosion along with the geographical location using a Global
Positioning System (GPS). A selection of the identified water soil erosions in the study
region is shown in Fig. 1.
We translated the observations of the field survey into maps of non-degraded and
degraded areas. These areas were plotted in an inventory map and prepared for further
analysis, although not all desert areas are fully covered by the survey.

**2.1.2 Dust aerosol map**
The large desert area to be covered is a motivation for the use of satellite data for
estimating dust sources. We used MODIS images from the Terra (morning) and Aqua
(afternoon) satellites (Vickery and Eckardt, 2013) to identify dust aerosols. We define
dusty days, when the horizontal visibility is less than 2000 m for at least one hour during
the day based on available weather stations in Iran (Vickery and Eckardt, 2013;
Boroughani et al., 2021). According to the mentioned condition, more than 500 dusty
days were identified during 2010–2021 distributed over the stations in Birjand,
Zahedan, Kerman, Bam, Doostabad, Bisheh, Rafsanjan and Mighan. We pair the station
observations with satellite data to estimate the spatial extent of the dust aerosol plumes.
Due to the overpass of the Terra and Aqua satellites once per day, we acquired 28
satellite images from the MODIS sensor that during times when the weather stations
had documented dusty conditions in the ten year period. For identifying pixels with dust
aerosols in these images, we calculate four different dust indices (BTD2931, BTD3132,
NDDI and D) for dust aerosol identification.(Boroughani et al., 2020, 2021
Hahnenberger and Nicoll, 2014).
$$B(T, \lambda) = \frac{2hc^2}{\lambda^5 \frac{hc}{(e^{\lambda kt}-1)}} \tag{1}$$
where B(T, λ) represents the Planck equation at λ (µm), T is the BT (K), h is the
Planck's constant ($6.626 \times 10^{-34}$ m2kgs$^{-1}$), k is the Boltzmann's constant ($1.38 \times 10^{-23}$)[5], c
is the speed of light ($2.99 \times 10^8$ ms$^{-1}$), and T is the temperature (Hao et al., 2007)

$$T = \frac{hc}{\lambda k ln(1 + \frac{2hc^2}{L\lambda^5})} \tag{2}$$
Using Planck's equation, the value of the temperature can be derived, where L is the
amount of radiance in the images (in Wm$^{-2}$sr$^{-1}$µm$^{-1}$).



$NDDI = (p_{2.13} - p_{0.469})/(p_{2.13} + p_{0.469})$

155    (3)


where $\square_{2.13}$ and $\square_{0.469}$ depict the reflectance value at the top-of-atmosphere at 2.13
and 0.469 μm, respectively (Qu et al., 2006)

$D = exp\{-[rr \times a + (BTD - b)]\}$         (4)
where rr shows the reflectance proportion among wavelengths of 0.54 μm and 0.86 μm
and BTD is the difference among the bands 11 and 12 μm; a and b are constants took
during the initial calibration (Eq. 1). (Qu et al., 2006; Miller, 2003; Hao et al., 2007;
Boroughani et al., 2020, 2021).
We compute false color maps using four combinations of channels (1: NDDI, B4, B3;
2: D, BTD2931, NDDI; 3: D, BTD3132, NDDI; and 4: BTD2931, B4, B3) in ENVI
software. We choose these four different indices for cross-validating the presence of
dust aerosols. With each of these methods we see dust aerosol in different color and
quality in the MODIS images on the 28 days. After combining the four methods in the
software ENVI, we choose the method that shows the dust plume in the MODIS image
more clearly as the best method (Boroughani et al., 2020, 2022). This method is based
on a cone of dust diffusion seen in the processed MODIS images, where the apex
denotes the dust's source (Lee et al., 2009; Walker et al., 2009). Ultimately, the
inventory map of the dust aerosols in the Lut watershed was created.

**2.2. Identification of key factors controlling for aeolian and water erosion**
To develop DSSA and SESA, the identification and selection of appropriate dust
sources and soil erosion effective factors is necessary. The main factors affecting DSSA
and SESA were selected and constructed based on literature, available data and
geographical maps (Torabi et al., 2021; Zabihi et al., 2018; Boroughani et al., 2020;
Gholami et al., 2020a). The considered factors in this study included: elevation, land
use, slope of terrain, lithology, annual rainfall, distance from rivers, distance from
roads, the Topographic Witness Index (TWI), and Normalized Difference Vegetation
Index (NDVI). Various sources were used to gather data for these factors, introduced
in the following in more detail. All collected data were mapped to a horizontal grid of
1km resolution.



The shuttle radar topography mission (SRTM) images were used to create the digital
elevation model (DEM, , Fig 3c) (Ghorbanzadeh et al., 2018). The lowest and highest
elevation of the study area is 124 m in the centre of the desert and 3966 m at the western
and eastern margins of the study watershed, respectively (Fig. 3c). Vegetation cover
considerably supports soil conservation. Areas with low vegetation cover would be
more sensitive to both erosion by water and wind (Arabameri et al., 2019a; Gholami et
al. 2019b). Therefore, we use the Normalized Difference Vegetation Index (NDVI) to
assess the vegetation cover in the study area from MODIS images following
(Arabameri et al., 2019a; Boroughani et al., 2020):
$\text{NDVI} = \frac{NIR+R}{NIR-R}$
Where R is the red (0.620-0.670 µm) and NIR is near-infrared bands (0.841-0.876 µm)
(Fig. 3d).
Annual rainfall (Fig. 3e) were obtained from Iran Meteorological Organization for the
period of 2000-2021. Mean annual rainfall were calculated using 40 different
meteorological stations located within or close to the watershed (Fig.3e). The inverse
distance weighting (IDW) interpolation method was applied to integrate rainfall over
the study area in the ArcGIS environment (Gholami et al., 2020a). Topographic
Wetness Index (TWI), which indicates the spatial distribution of areas of potential soil
saturation, is an effective factor to indicate water erosion including landslides and also
flooding (Arabameri et al., 2019b). TWI which determines the dry and wet zones
calculated as (Beven and Kirkby 1979):
$TWI = ln(\frac{\alpha}{tan\beta})$
where α is the cumulative up-slope area from a point (per unit contour length) and β is
the slope angle at that point. This index was calculated in the SAGA-GIS environment
and classified into four groups viz. 14-17, 17-19, 17-21, 21-24 (Fig. 3f). The aspect
map was also generated using DEM and grouped into ten classes (Fig. 3 g). Distance
from road is an indicator of infrastructure development which influences soil erosion
and land degradation (Torabi et al., 2021). This factor is shown in five classes in Fig. 3
h. Distance from river is one of the most effective factors on water-caused erosion
(Amiri et al., 2019) which is classified into six groups (Fig. 3i).





The slope map (%) was created using a Digital Elevation Map (DEM, Fig. j) and
classified into five groups including 0-3%, 3-6%, 6-12%, 12-21%, and 21-54%. The
lithology map indicates eleven different soil classes in the study area (Fig. 3k).
Land use and soil maps were obtained from base maps developed by the Iranian Forest,
Rangeland, and Watershed Management Organization (https://frw.ir/). In the study
region, there are fourteen land-use classes including wetlands, rangelands of three states
(poor, medium, and rich), dry farming, agricultural lands, urban area, fallow land, rock-
covered land, wetland, saltland, woodland, bare surfaces, and sand dunes (Fig. 3m). A
large percentage (83%) of the watershed area is covered by bare land, poor rangeland,
and sand dunes. All three land use classes are prone to wind erosion due to sparse or no
vegetation.

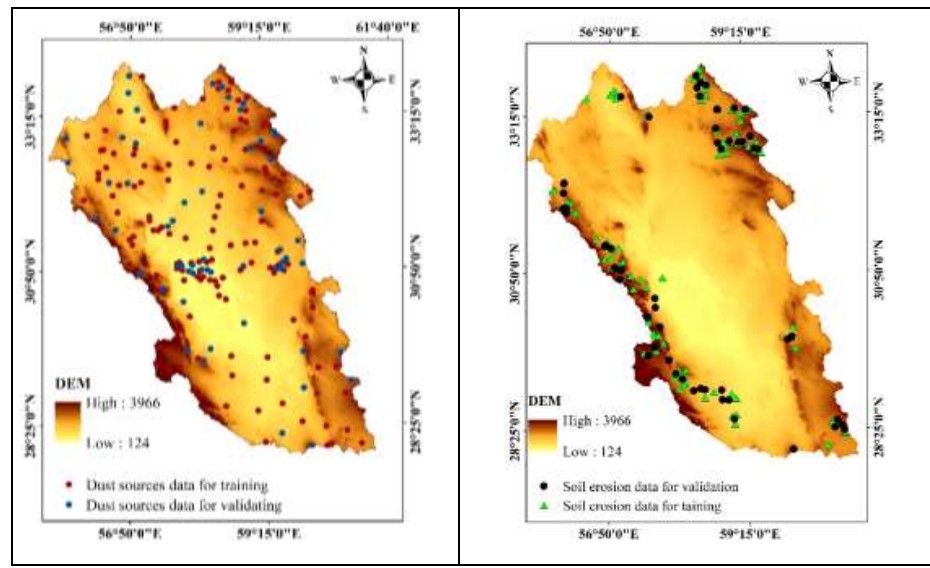



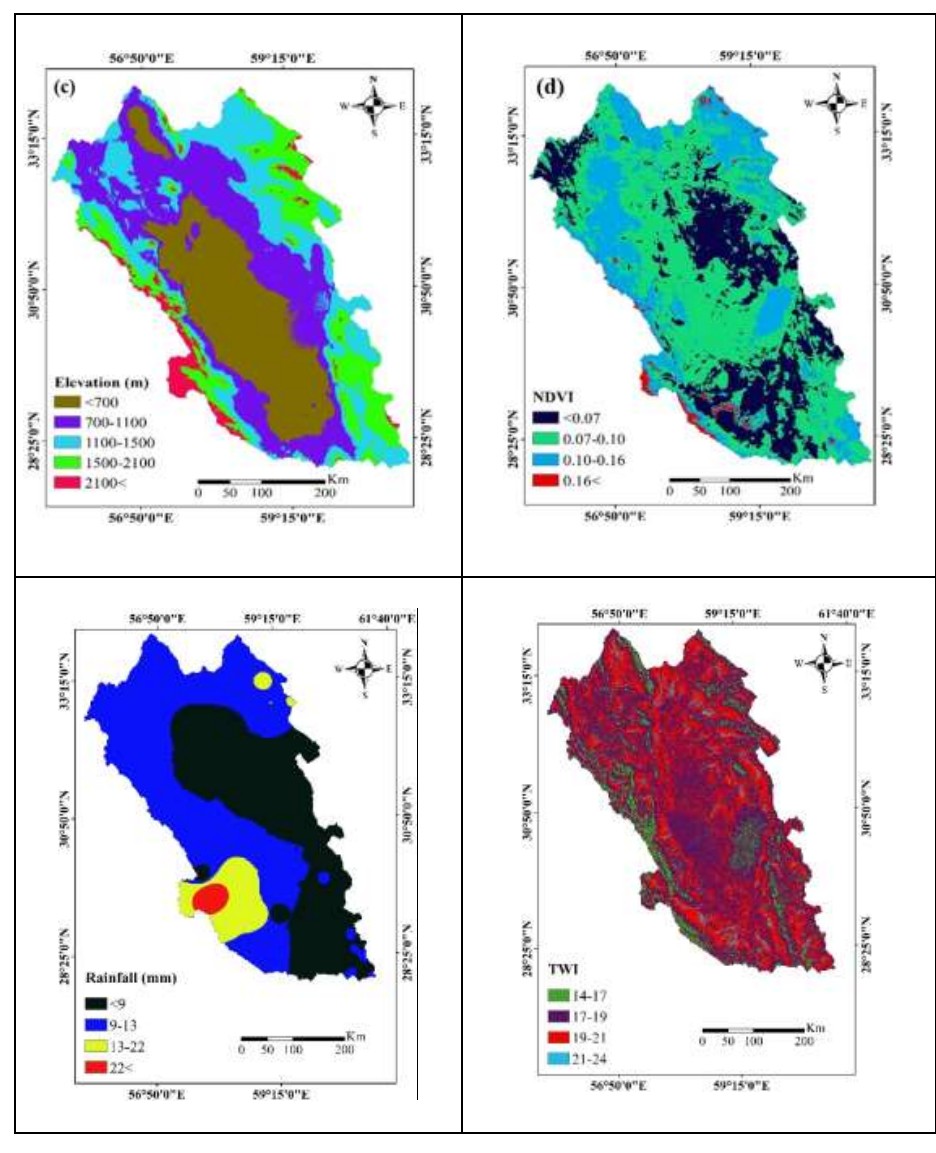



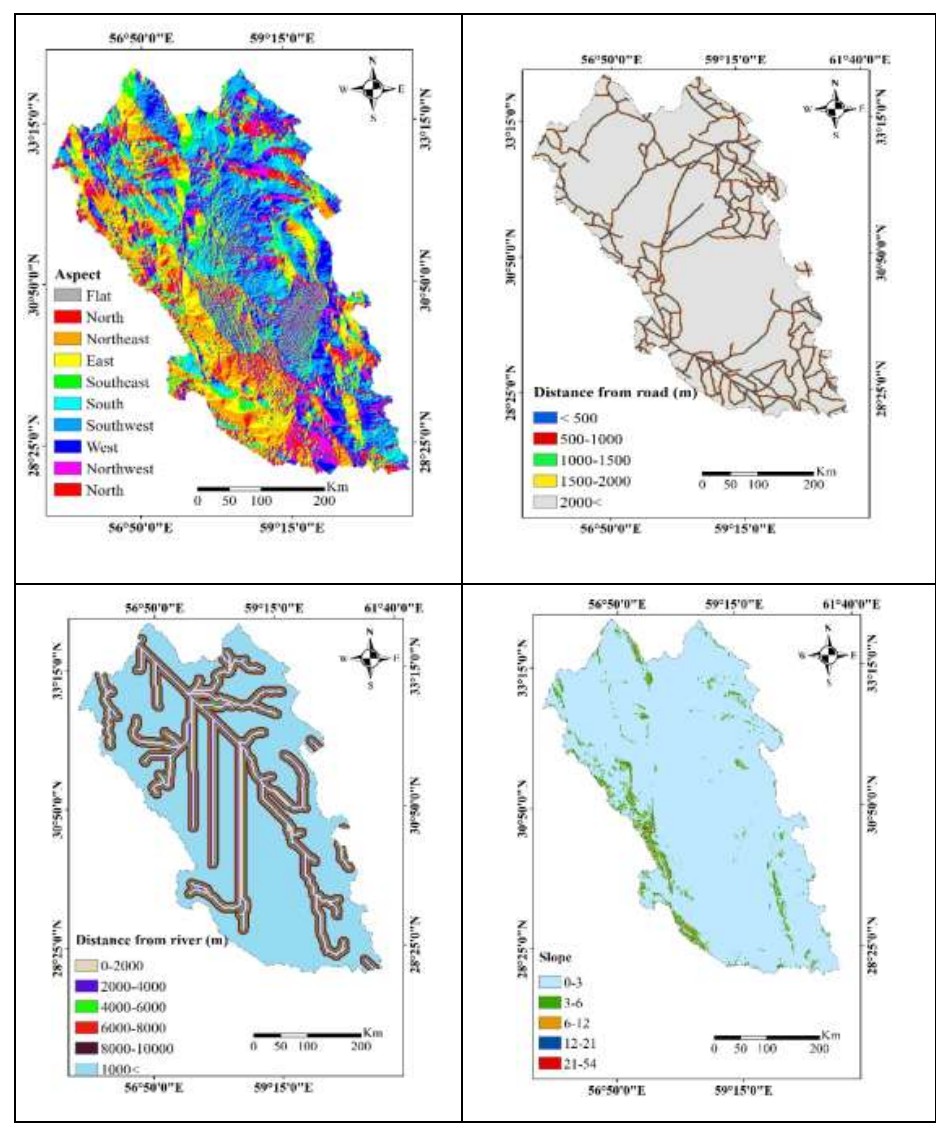

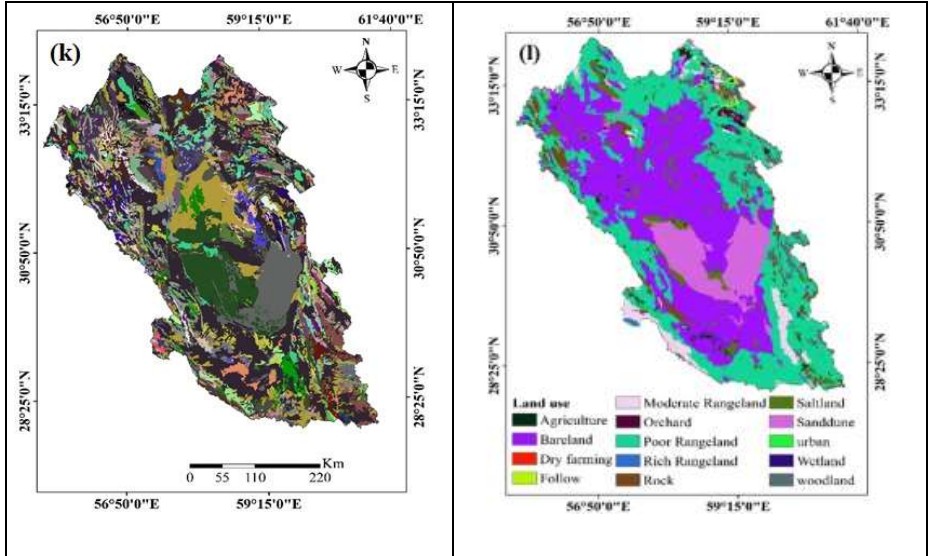

Fig.3 Location of dust observation points for training and validation (a), water-induced soil erosion
points for training and validation (b), and the conditional factors (Elevation (c), NDVI (d), Rainfall (e),
TWI (f), Aspect (g), Distance from road (h), Distance from river (i), Slope (j), Lithology (k), Land use

232                                         (l)) in the watershed.


**2.4. Spatial mapping of DSSA and SESA using machine learning algorithms**
We combine the two susceptibility maps for DSSA and SESA to create the land
degradation hazard map with regards to water- and wind-induced soil erosion. For both
types of soil erosion, three machine learning models were constructed and applied. The
land degradation susceptibility map was then created by synthesizing the results for
both soil erosion types in an ArcGIS 10.5 environment, and the land degradation
susceptibility was ultimately evaluated with four classes.
A wide range of machine learning algorithms has been applied for spatial mapping of
environmental phenomena in the past. The effective factors described in Section 2.2
and the inventory maps of water and wind erosion were used as the input of the machine
learning algorithms. In the present study, the algorithms of random forest (RF), artificial
neural network (ANN), and flexible discriminate analyses (FDA) were used to produce
DSSA and SESA maps. We choose three different algorithms to test the dependency of
the results on the method as a measure of uncertainty. The  three algorithms are
described in more detail in the following.



### 2.4.1 Random forest (RF)

Random forest developed by Breiman (2001) is a machine learning algorithm for non-
parametric multivariate classification. RF builds multiple trees using a random
selection of the training dataset. The data not included are called out-of- bag (OOB)
determines the model accuracy using generalization error estimation (Breiman 2001).
Diversity among the classification trees increases using resampling the data with
replacement and also randomly change of predictors set during tree induction processes
(Youssef et al., 2016). Information from numerous decision trees has been combined in
the RF algorithm.
Generally, it is essential to define two parameters to run the RF model including the
number of trees (ntree) and the number of factors prepared from the data shown in Fig.
3 (mtry). The former are built while the RF model is running, while the latter is used in
the tree-building process. Both the number of trees and factors need to be optimized to
minimize the generalization error (Rahmati et al. 2016). The optimisation was done
through sensitivity tests.

### 2.4.2 Artificial neural network (ANN)

The artificial neural network (ANN) is a machine learning tool developed by imitating
human brain performances and making connections between inputs and outputs
(Sakizadeh et al. 2017). The human brain is mimicked in two ways: Firstly, obtaining
information and knowledge using a learning process, and secondly, storing knowledge
using synaptic weights. Therefore, ANN has been identified as the model that finds the
optimal solution for non-linear problems, such as dust source and soil erosion
susceptibility, by identifying patterns with conditioning factors (Ghorbanzadeh et al.
2019). In an ANN, a neuron is the smallest data processing unit which could make many
neural network structures and be used in research for different purposes. The standard
structure of ANN consists of three layers, namely, the input layer, hidden layers, and
the output layer. The input layer consists of training data and conditioning factors of
dust source, the neurons in the hidden layer analyze the complex information contained
in the data, and the output layer are the maps of dust source susceptibility. In this
structure, the neurons across the same layer are not connected, but they are linked with
neurons in the previous and subsequent layers. In ANN, the algorithm determines a
weight for each input factor and a transfer function to build results (Kalantar et al.
283 2017).




### 2.4.3 Flexible discriminate analyses (FDA)

The modification of the linear regression model for the application to non-linear problems is the purpose of FDA (Avand et al. 2021). Nonparametric regression models, nonlinear discriminant analysis, and classification methods are combined into one framework. This algorithm is flexible for non-linear classifications because non-linear transformation is used and clusters are soft (Kalantar et al. 2020), here clusters for the relationship between soil erosion and the predictor factors from Fig. 3. In this way, variables in FDA are firstly aligned with the multivariate adaptive regression splines (MARS) and then dimension reduction is performed (Kim and Kim 2021). FDA can overcome the problem of linear discriminant analysis (LDA) and it is minimizing the square average of the residuals (Mosavi et al. 2020), while linear regression is replaced by nonparametric regression in FDA. Therefore, FDA has the potential to apply for non-linear natural problems such as soil erosion, dust, flood, and landslide.

### 2.5. Evaluation of machine learning algorithms

In our DSSA and SESA assessment, 70% of point data are randomly selected for the training dataset and 30% for model validation. The prediction accuracy of the machine learning algorithms is assessed by comparing the DSSA map with the validation dataset of dust sources. These data were extracted from MODIS images and some indicators which was explained in section 2.1.2. The Receiver Operating Characteristic (ROC) curve and the Area Under the Curve (AUC) are applied following past studies that used these to test the prediction skill of a model for the occurrence or non-occurrence of the studied phenomena (Naghibi et al. 2017). The AUC ranges from 0 to 1 in which the models that better perform represent the AUC close to one.

### 3. Results and Discussion

### 3.1. Spatial distribution of DSSA

### 3.1.1. Dust aerosol detection

An illustration of a dust storm seen in MODIS FCC satellite imagery over the Lut watershed on August 7, 2019, is shown in Fig. 4. Following a visual analysis of the images, we determined that the false colour combination (R: BTD2931, G: Band 4, B:



Band 3) is the best and applied it to 26 MODIS images of dusty days. As a result, the
Lut watershed's dust source locations were identified (Fig. 4).

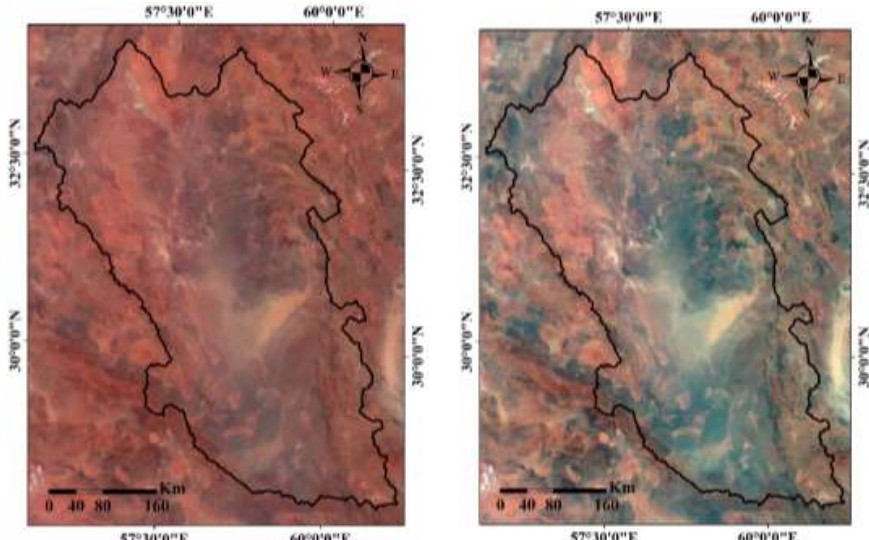

Fig.4 The dust storm on 07 August 2019, as seen above is an example of the visual
inspection of a dust storm (a) MODIS true colour (Red: Band 5, Green: Band 4, Blue:
Band 3), and (b) enhanced MODIS satellite photos,  (Red: BTD2931, Green: Band 4,

322                                    Blue: Band 3).


**324     3.1.2 The importance of conditioning factors for DSSA**

Since multicollinearity among factors has been identified as an obstacle to explaining
the results (Roy and Saha 2019), the Variance Inflation Factor (VIF) was calculated to
assess the relationships among conditioning factors. This was conducted because
multicollinearity among factors will decline the accuracy of the models (Arabameri et
al. 2019b). In the present study, VIF values for DSSA mapping range from 1.05 to 1.57
which illustrated no collinearity among the eight factors. Therefore, no exclusion was
applied and all factors were considered in successor calculations and modeling.
The importance and impact of each factor depend on the machine learning algorithms.
The result of DSSA mapping using RF showed that NDVI, elevation, land use, and
lithology had the greatest degree of effect among conditioning factors. In addition, the
impacts of these factors on DSSA have been proved by previous investigations





(Gholami et al. 2020a, 2020b). Other factors such as the distance from rivers, lithology,
rainfall, and slope were identified as rather weak predictors, respectively. These
findings agree with other research (Boroughani and Pourhashemi 2020, Darvand et al.
339    2021).

The FDA approach showed that however elevation, NDVI, and land use had the highest
effects on dust sources susceptibility, other factors had no impact on DSSA. Similarly,
with ANN, elevation, NDVI, and land use were identified as the three most effective
factors, and other factors were weaker predictors rather than formers. However these
two models of FDA and ANN provide similar results in term of the importance of
conditioning factors, FDA could be used rather than ANN because of its higher
accuracy which is shown in the next section.

**3. 1. 3 Spatial distribution of dust source susceptibility**

The dust source susceptibility (DSS) maps created by RF, FDA, and ANN are classified
into five risk classes (very high, high, moderate, low, and very low) shown in Fig. 5.
These classes are set as in earlier studies (Mosavi et al., 2020; Boroughani,
Mohammadi, Mirchooli, & Fiedler, 2022). The results of the model evaluation using
ROC indicates that the RF model with an accuracy of 75.0% provides the most accurate
outputs. FDA and ANN had similar performances with the accuracy of 71.7% and
70.7%. In terms of True Skill Statistic (TSS), similar results have been obtained in
which RF with accuracy of 45.8% had again the best performance in comparison to
FDA (32.4%) and ANN (35.8%). In this way, RF introduces different priorities for the
effective factors in comparison with FDA and ANN. RF proposes NDVI, elevation,
land use, and lithology as the most important factors, while FDA and ANN suggest
elevation, NDVI, and land use as the most influencing factors. The dominance of
NDVI, elevation and land use as the most effective factors for DSS is consistent with
the understanding of dust source locations that are typically found in topographic
depressions with sparse or no vegetation. The DSSA map from RF was selected for
further analysis due to the highest accuracy, although the differences to FDA and ANN
are in the statistical sense relatively small. According to the DSSA maps, 29% and 17%
of the watershed were classified as areas of high and very high DSSA, i.e., almost half
of the study area. Only 4% and 16% of the watershed have a very low and low
susceptibility to soil erosion through winds, respectively. The spatial extent of high and
very high risk areas from RF is smaller than the ones obtained by ANN and FDA. These



results are consistent with other research, indicating that RF allows more detailed
spatial mapping of dust source susceptibility compared to other machine learning
algorithms (Rahmati et al. 2020, Gholami et al. 2019b, Darvand et al. 2021).

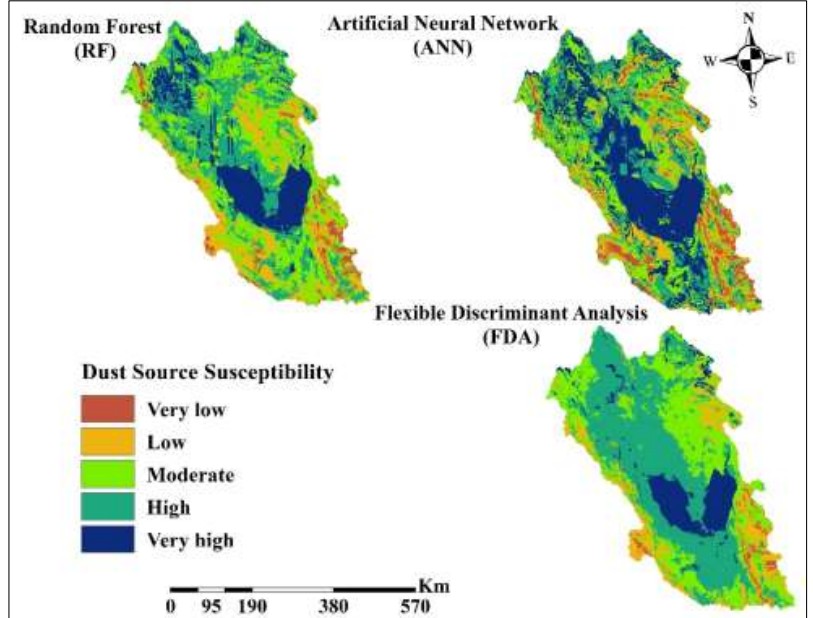

Fig. 5 Dust sources susceptibility area (DSSA) based on random forest (RF), artificial neural network
(ANN), and flexible discriminate analyses (FDA)

**3.2. Soil erosion susceptibility map**
**3.2.1 Relative influential conditioning factors for SESA**
For SESA, RF indicates that rainfall, TWI, slope, elevation, land use, and geology are
the most important conditioning factors. Distance from roads and rivers was recognized
as the least important factors. These findings of the impact of conditioning factors for
SESA are similar in other regions (Arabameri et al. 2019a, Hosseinalizadeh et al. 2019).
For ANN, TWI, slope, and land use were the most effective factors for prediction which
is followed by NDVI, land use, and distance from the river. The results from FDA
indicated that the most important conditioning factors are TWI, slope, and elevation,
geology, and NDVI.
A large area of the watershed is land with typically little rain and vegetation cover such
that bare soil is the main physical attribute in the watershed. This kind of surface is





known to be prone to water-induced soil erosion, when rain events occur. The erosion
can be particularly pronounced over slopes. This understanding is consistent with all
algorithms pointing to a major role of TWI and slope for SESA.
Some environmental factors (rainfall, TWI, slope, elevation, and geology) influence
SESA more than DSSA. Land use as a human-induced conditioning factor, however,
affects both SESA and DSSA, which underlines the importance of land-use planning
and management.

**3.2.2. Spatial modeling of SESA**
Fig. 6 shows the SESA predictions from the three machine learning algorithms,
classified by the soil erosion risk in the ArcGIS environment. Validation of the three
machine learning algorithms highlights that RF was again the most reliable algorithm
amongst the three, indicated by the best prediction rate. Based on ROC, RF yields a
94% accuracy for SESA (Fig. 6c). The ROC coefficient of ANN and FDA were slightly
lower, but still high with an accuracy of 91% and 89%, respectively. In the case of the
TSS index, better performance was obtained again for RF (89%) rather than ANN
(78%) and FDA (78%).
The majority of the land in the watershed (81%) has a high and very high risk for water-
induced soil erosion by RF. This is slightly lower than for ANN and FDA which
classified 85% and 89% of the watershed as high and very high susceptible areas. The
high and very high susceptible areas for water-driven soil erosion are mostly located in
the north and south-west parts of the watershed. The high and very high susceptible
areas have socio-economic implications, particularly because most settlements and
cities of the watershed are located in the same regions. This can mean that human
activity is a contributing factor for the water-induced soil erosion.



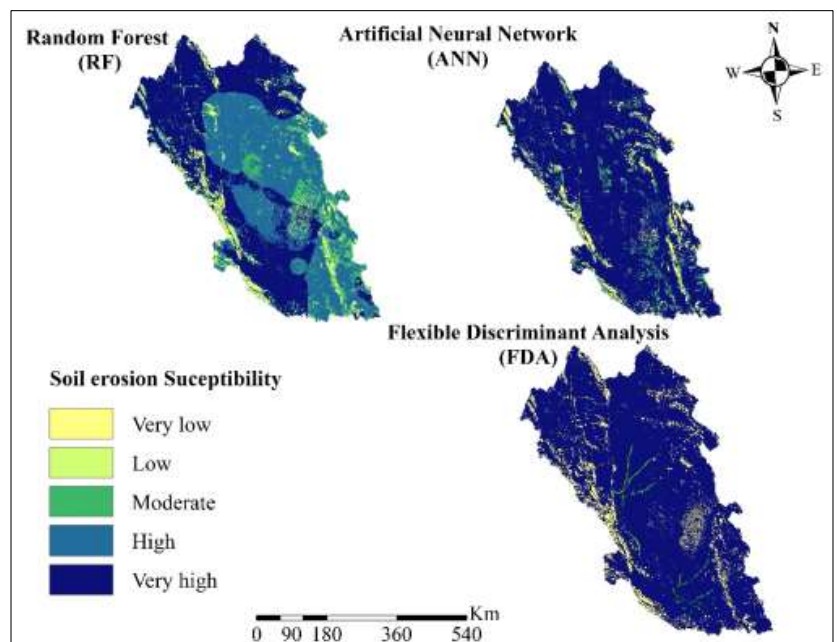

Fig. 6 soil erosion susceptibility areas map (GESM) using random forest (RF), artificial neural network
(ANN), and flexible discriminate analyses (FDA)

**3.3. Land degradation susceptibility**

The majority of the study watershed is susceptible to a substantial risk for land
degradation. The spatial distribution of land degradation susceptibility, shown in Fig.
7, indicates that only 4% of the land area has low to very low risks of land degradation.
Areas susceptible to both soil erosion by water and winds together constitute 43% of
the total area. Approximately 45% and 8% of the study area are at risk to soil erosion
by water and wind, respectively. Taken together, it means that the majority of the Lut
watershed falls under the category of land degradation risks. The watershed accounts
for 12.5% of the total land of Iran. The findings of the present study is therefore
consistent with a report that indicated water erosion as an environmental hazard in Iran
(Bui et al. 2019).

The areas that falls under the category of both kind of land degradation might be most
vulnerable concerning local self-sufficiency for food security and sustainability of
human activities. For instance, dust storms drive water loss through failure of
agricultural crops in Iran (Boroughani et al. 2022). Moreover, the adverse impacts of




water-induced soil erosion is known from numerous other regions (Lal and
Moldenhauer 2008, Gao et al. 2015, Standardi et al. 2018).

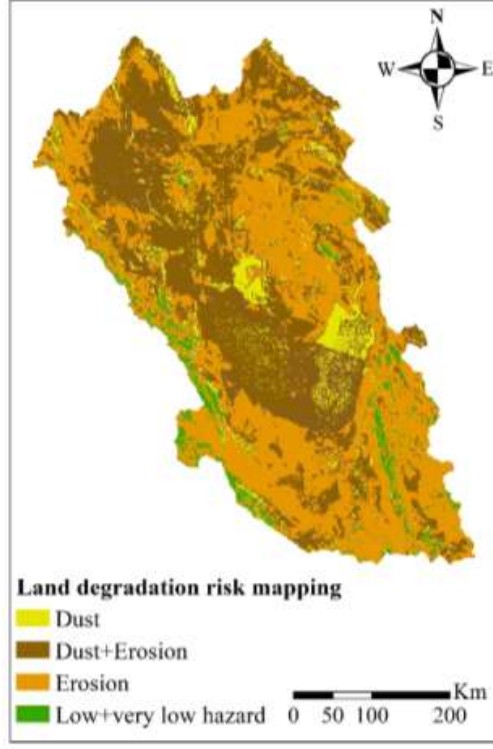

438           Fig. 7 Land degradation susceptibility map in terms of soil erosion and dust sources areas

**Conclusion**
Investigation of soil erosion through water along with wind-driven soil erosion from
dust sources have received little attention in past studies, despite their importance for
land degradation with associated social, economic, and environmental impacts. The
present study used several different data sets, conducted a field survey  and paired the
data with three different machine learning algorithms to construct spatial maps for areas
of risk for land degradation for the Lut watershed in Iran. Three machine learning
algorithms were successfully applied to create land susceptibility maps describing dust
aerosol occurrence considering methodological uncertainty. In addition, these models
were used to identify the areas prone to soil erosion by surface water runoff. These
obtained maps were synthesized to generate a single map for risks of land degradation.



The results of the present study show that the random forest algorithm outperformed
the other two machine learning approaches for both dust sources and soil erosion
susceptibility mapping with an accuracy of 75% and 94%, respectively.
As expected, the vegetation cover, terrain elevation, land use, and geology were
important prerequisites for dust-emission occurrence in the watershed, while rainfall,
Topographical Wetness Index (TWI), terrain slope, terrain elevation, land use, and
geology were identified as the most influential factors for water-induced soil erosion.
Based on the land degradation map, almost the entire study region is at risk. A large
fraction of 43% of the area is prone to both high wind-driven plus water-driven soil
erosion. In addition to these areas, another 45% and 8% of the area have a risk for water-
driven and wind-driven soil erosion, respectively. These results can potentially be
useful for managers and policy makers to identify local hotspots for land degradation
to implement mitigation and adaptation measures in this watershed. Future studies
could work on improving the spatial resolution and coverage of the risk assessment for
providing more information on risks for land degradation. It requires more
measurements for soil erosion by water and winds to train the machine learning models.

**Acknowledgement**
SF acknowledges funding from the German Research Foundation (DFG) for SFB
1502/1–2022 (Project: 450058266).

**Conflict of Interest**
The authors declare that there is no conflict of interests regarding the publication of
this article.

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
