# Peer review of "Mapping land degradation risk due to wind and water erosion"

_EGUsphere, 2022_

## Referee Comment (RC3)

The study is about "Mapping land degradation risk due to wind and water erosion". Principles and methodology are well supported. However, I would like to consider some specific comments below as minor revision. In general, there are some grammar errors throughout the manuscript. Please check the spellchecking in addition to these minor issues.

Page 5: Only 23 dusty days are used ….. out of 500 dust days, I speculate…every year: 28 dusty days (2000-2020). I would like to see a histogram with MODIS AOD (500nm, 2000-2020) at least for one key dust station. I think all days with AOD>0.5 (500nm) are dusty days. Is the use of 23 days really a representative approach?

Why not showing a correlation with wind speed vs AOD for several selected observational sites? Your argumentation is not very much convincing without some demonstrations (visualizations). One could get the impression, the work is based, at the end, on a few 'beauty data' only, but does not allow solid conclusions and does not provide insight into the complex problematic.

Did you consider these days with visibility less than 2 km as non fog/haze days? How did you differentiate between fog and dust?

Once per day cannot be considered as high resolution?

Considering that the map of dust sources for Iran was prepared by the Geological Organization, why did you use satellite images to produce it? How accurate is your generated map compared to the ground map?

How to deal with the satellite data with different spatial resolutions in the study?

Line 26: **The** water and aeolian soil erosion **maps**

Line 32: the risk of land degradation **in** an inhabited region

Line 43: soil erosion in **a** short time

Line 49: has detrimental impacts **on** the Earth system

Line 52: therefore necessary **for** developing a better understanding

Line 85: to the increasing dust concentration **in** southwest Asia

Line 118: That information **is** extracted from data collected during an own field **survey** paired with a previous research (delete **"a"** before previous)

Line 141: in the ten**-**year period (add hyphen for "ten**-**year")

Line 162-163: constants **taken** during the initial calibration

Line 168: we see dust aerosol in different color**s** and quali**ties** in the MODIS images **over** 28 days

Line 177-178: the identification and selection of appropriate dust sources and soil erosion-effective factors **are** necessary.

Line 183: the Topographic W**e**tness Index (TWI), (Not Witness)

Line 199: Annual rainfall (Fig. 3e) **was** obtained from

Line 195: Mean annual rainfall **was** calculated using

Line 261:  The former **is** built while the RF model

Line 276: three layers, namely, the input layer, **the** hidden layers

Line 279: and the output layer **is** the maps of

Line 304: and some indicators which **were** explained in section 2.1.2

Line 356: similar results have been obtained in which RF with **an** accuracy of 45.8%

Line 364: although the differences **between** FDA and ANN are in the statistical sense relatively small.

Line 381: Distance from roads and rivers **were** recognized as the least important factors

Line 414: human activity is a contributing factor **to** the water-induced soil erosion.

Line 425: the study area are at risk **of** soil erosion

Line 428: The findings of the present study **are** therefore

Line 431: The areas that fall under the category of both kind**s** ("falls" should change into "fall")

Line 435: the adverse impacts of water-induced soil erosion **are** known

---

## Author Comment (AC1)

**Mehdi Boroghani, Assistant Professor, Dr.**
*(Watershed Management Engineering)*
Research Center for Geoscience and Social Studies
Hakim Sabzevari University
Sabzevar, IRAN
E-mail: m.broghani@hsu.ac.ir

**Dear Prof. Dr. Lorena Grabowski**

**Editor-in-Chief**
**SOIL**

*23 March 2023*

Greetings;

       Attached please find the revised version of our manuscript entitled "**Mapping land degradation risk due to wind and water erosion**". Full revision has been carried out by responding to the comments and considering suggestions made by the reviewers. The implementation of those valuable comments and suggestions has significantly improved the quality of the paper for which we are grateful to the editor and the reviewers. I am also attaching a note in which major changes carried out in the paper have been explained. All revised parts have been highlighted in red color.

       I hope the emendations caused to consent the respected editor and make my paper well qualified for further processing leading to final acceptance and publication. The acknowledge receipt of the same and informing me about the final status of the paper is appreciated.

Sincerely,

Dr. M. Boroghani
Enclosure

<h2 style="text-align:center; color:blue">Reviewer 1 # Revision comments:</h2>

The paper entitled as "Mapping land degradation risk due to soil erosion by storms and water:

The authors use machine learning to assess wind and water erosion susceptibility in the study region. The article is interesting and has potential. I carefully read the manuscript and it is suitable for publication after minor revision.

**Ans.** Thanking you so much for your encouraging and nice words towards my work.

1. I suggest some literature that can be helpful in this regard. But authors are not bound to cite them in this manuscript:

   https://doi.org/10.1002/ldr.4380

   https://doi.org/10.1002/ecs2.2650

**Ans.** It was added as the respectful reviewer suggested.

2. MODIS first time used, abbreviation not explained and no reference.

**Ans.** It was revised. MODIS is the abbreviation of "Moderate Resolution Imaging Spectroradiometer" which is added in Line 61.

Line 66**:** The present study uses Moderate Resolution Imaging Spectroradiometer (MODIS) satellite images to detect dust aerosols over the Lut Desert.

3. Lines 77-79: Please state the possible applications of your study

**Ans.** It was added.

Line 482: The results of the study will be helpful and applicable for identifying water-induced and dust sources hotspots across the watershed and prioritizing appropriate conservation measurements and rehabilitative policies.

4. Please indicate the limitations of this work.

**Ans.** It was added.

Line 518-521: The current study has some limitation including the small sample size and non-uniform distribution of water-induced soil erosion points because of lack of accessibility to a road network in some parts of the watershed.

5. Dust sources are locations of dust particles where are susceptible to wind erosion. So I think that "wind erosion susceptibility" is more appropriate than dust source susceptibility (DSS).

**Ans.** The authors agree with the respectful reviewer that dust sources are the location of dust particles. But this term has been also used by previous research (such as Identifying sources of dust aerosol using a new framework based on remote remote sensing and modelling; Spatial mapping of the provenance of storm dust: Application of data mining and ensemble modelling) which conducted using similar methods. Therefore, it would be better to remain this term in the present study to avoid any confusion. Correspondingly, the title of the study is changed to "Mapping land degradation risk due to land susceptibility to dust emission and water erosion"

6. What is the outlook or scope of the future research for this work?

**Ans.** However there is suggested for the future research in the conclusion of the manuscript, another suggestion was provided and added to this section as below:

Line 523-531: Future studies could work on improving the spatial resolution and coverage of the risk assessment for providing more information on risks for land degradation. In addition, it is suggested that future research should estimate the role of other climatic factors such as humidity, and air temperature on soil erosion and dust source susceptibility. Prediction of NDVI and rainfall as the most effective factors on soil erosion and dust sources and estimation of their impacts on future water induced-soil erosion and dust sources susceptibility are also suggested for the other studies.

---

## Author Comment (AC2)

**Mehdi Boroghani, Assistant Professor, Dr.**
*(Watershed Management Engineering)*
Research Center for Geoscience and Social Studies
Hakim Sabzevari University
Sabzevar, IRAN
E-mail: m.broghani@hsu.ac.ir

**Dear Prof. Dr. Lorena Grabowski**
**Editor-in-Chief**
**SOIL**

Greetings;

Attached please find the revised version of our manuscript entitled "**Mapping land degradation risk due to wind and water erosion**". Full revision has been carried out by responding to the comments and considering suggestions made by the reviewers. The implementation of those valuable comments and suggestions has significantly improved the quality of the paper for which we are grateful to the editor and the reviewers. I am also attaching a note in which major changes carried out in the paper have been explained. All revised parts have been highlighted in brown color.

I hope the emendations caused to consent the respected editor and make my paper well qualified for further processing leading to final acceptance and publication. The acknowledge receipt of the same and informing me about the final status of the paper is appreciated.
Sincerely,

Dr. M. Boroghani
Enclosure

Comment on egusphere-2022-1511
Anonymous **Referee #2**
Referee comment on "Mapping land degradation risk due to wind and water erosion" by Mahdi Boroughani et al., EGUsphere, https://doi.org/10.5194/egusphere-2022-1511-RC2, 2023
The manuscript "Mapping land degradation risk due to win and water erosion" presents an interesting research, with significant data and results. However, I consider that in its present form the manuscript cannot be published and major revisions are needed before a new submission.
**Ans:** Thanks for reading this manuscript and your comments. I hope the revised version would have fulfilled your scientific expectations.

I do not agree with the sentence that land degradation receives little attention. I think that this topic has been widely and worldwide study with different studies, methods

**Ans:** We agree with you that land degradation has received much attention in the whole world, but we mean the simultaneous investigation of water and wind erosion. We revised in the manuscript line 16

Lines 32-34. The last sentences of the abstract should be also improved. Why larger area? Is it necessary for land management? These ideas should be better developed.

**Ans:** Due to the fact that dust affects large areas and the place of the source to transportation and sedimentation is very wide, therefore, the larger the area, the better the investigation is done. And in the continue, land management and better ideas will be developed.

In general, I consider that the introduction section should be improved. My suggestion is to present first the topic of land degradation (better developed), second dust-wind erosion, water erosion and finally all the ideas related to the methodology (machine learning…). In addition, the objectives should be improved and I consider that a research hypothesis should be included.

**Ans:** The introduction section was revised as suggestion. Objectives and hypotheses were revised and added in line 86-95.

Line 86-95: This research is conducted to test some hypotheses including (1) the central and western parts of the watershed are the highest susceptible areas to water erosion and aerosol emission, respectively (2) NDVI and land use are the most important factors for water erosion and aolian emission and (3) Central areas are the most prone parts of the watershed to these phonemona. Correspondigly, the aims of the current study are (1) to assess the spatially resolved contribution of soil erosion by water and wind using three machine learning algorithms, (2) determine the most important factor influencing water and dust emission susceptibility and (3) to combine the findings into spatially resolved information on risks for land degradation and recognize the hotspot area in terms of water erosion and dust emission.

Likewise, the novelty of the manuscript and the international impact of the methods and results should be highlighted thought the text.

**Ans:** It was revised.

Line 81: However land susceptibility to soil erosion and dust emsission has been assessed in different and separate studies, it has attracted less attention to investigate both of them in the same study. So, the novelty of this study lies in constructing an integrated framework based on field survey, different environmental factors, and machine learning algorithms to assess both of water erosion and dust emission.

Line 408-411: As mentioned before, the watershed is one of the key regions with dust concentration in southwest Asia. Spatial distribution of dust sources in this region is a key roadmap for preventive and adaptive measurement. This would reduce dust emission across the watershed, region, and even other near countries.

Line 464-466: Mutually, intensified soil erosion might lead to migration of resident people to other places and even other countries.

I consider that lines 88-93 should be moved to the introduction section, as it presents a research problem.
**Ans:** It was revised. A part of that was moved to introduction.

You should also improve the study area description (altitude, slopes, climate, soils…) a brief presentation of the study area.
**Ans:** It was revised.
Line 100-107: This watershed include a great diversity of topographic characteristics, with an elevation ranging from 124 to 4269m, and slope ranging from 0 to 28.04 degree. In this region, southwest and northeast aspects have the most frequencies (34% of the area). This watershed covers some parts of the South Khorasan, Yazd, Kerman, and Sistan-Baluchestan Provinces of Iran. In addition, several important cities and towns such as Birjand, Tabas, Bam located in the watershed. Aridisols is the dominant soil order of the watershed in which it constitutes 40.1% of this region.

Line 117-127. This information should be better presented. I understand that you indicate a reference where the methods are explained, but the reader of this manuscript should know more about it. More information is needed.
**Ans:** It was revised.
Line 138-140: In the previous research, a combination of consulting with provincial experts, satellite images, recent aerial photos, and field survey were applied to identify soil erosion.

Figure 3. I think Figure 3 should be separated into several figures.
**Ans:** All of maps in this figure are related to the same procedure, and are used for machine learning algorithms. These maps were presented in one figure, because of ease of following the factors and avoiding any confusion. This format of presenting of controlling factors was also applied in different studies such as "Gully erosion zonation mapping using integrated geographically weighted regression with certainty factor and random forest models in GIS", " Comparative assessment using boosted regression trees, binary logistic regression, frequency ratio and numerical risk factor for gully erosion susceptibility modelling", and " A New Approach for Smart Soil Erosion Modelling : Integration of Empirical and Machine Learning Models"

Figure 3. Water-induced soil erosion points for training and validation. You should justify the location of these points, as they are limited-spatially.
**Ans:** However it is mentioned in line 143 that the field survey was conducted in accessible parts of the watershed, it is explained more as below:

Line 142-144: This field survey was carried out in accessible parts of the watershed in April 2020. These accessible parts are mostly distributed around the cities (such as Bam, Ravar, Shahdad, Baravar, Birjand, Tabas, etc) with proper road access located in the watershed.

Figure 3. Better legends should be included in some cases (rainfall, is it annual rainfall?, lithology?). In addition, the maps distance from roads and rivers present strange information.
**Ans:** Rainfall and Lithology were revised. Since there too many lithology classes in the study watershed, dominant classes are presented in this figure. Other classes would be illustrated in the appendix. The figures were also revised. Distance from road was double-checked, but there is the same information in other available layers.

Results, discussion. This section should be also improved, including international studies, comparisons, discussing your main results, your methodology…
**Ans:** It was revised.

Line 357-363: Land use and NDVI as an index of vegetation cover proved to have a controlling impact on wind erosion and dust emission (Gholami et al., 2020). Elevation is an effective factor for DSSA in which lowlands have higher impacts than highlands. This was confirmed by other studies such as Darvand et al., 2021. Lithology is another important factor in this watershed since dust emission is mostly occur in the sensitive lithology rather than resistant ones (Sissakian et al., 2013).

Line 398-399: In all three maps, it can be seen that the biggest potential for dust emission is located in the central parts (Lut Desert) of the watershed.
Line 415-425: There are some differences in the contributions of influential factors among models. So that, RF indicates that rainfall, TWI, slope, elevation, land use, and geology are the most important conditioning factors. Considering this watershed located in arid region of Iran, rainfall and TWI play decisive and crucial role in soil erosion among them. TWI which indicate soil moisture and water-saturated area (Silva et al., 2023) has been also identified an effective factor for different kinds of soil erosion such as rill-interrill, gully, and piping erosions (Sholagberu et al., 2017; Hosseinalizadeh et al., 2019). Slope influences also soil erosion rate through effecting on runoff velocity, vegetation cover, and soil type (Avand et al., 2022). This conditioning factor has been also reported as one of the most influential factor in most studies (Sholagberu et al., 2017; Pournader et al., 2018; Lei et al., 2020).
Line 431-433: TWI has an important impact on SESA in all three models. This is because the study watershed predominates with low slopes and elevations. The opposite result of this finding was obtained by Silva et al., 2023.
Line 452-456: High performance of RF model in classification issues is related to its potential to handle bigh datasets and aplly large number of conditioning factors (Naghibi et al., 2018). In addition, Rahmati et al., 2020 states that high accuracy of RF is the results of several advantage of this model such as iterative nature and preventing problems by overfitting (Rahmati et al., 2020).

As I suggest in the introduction section, the conclusion section should be also rethought, and improved.

**Ans:** It was revised.

Line 510-527: Based on the land degradation map, almost the entire study region is at risk. A large fraction of 43% of the area is prone to both high wind-driven plus water-driven soil erosion. In addition to these areas, another 45% and 8% of the area have a risk for water-driven and wind-driven soil erosion, respectively. The methods tested in this study could be later transferred to similar assessments in other regions around the world. Choosing this region in Iran is further motivated by the impact of land degradation on the country's economy. The current study has some limitation including the small sample size and non-uniform distribution of water-induced soil erosion points because of lack of accessibility to a road network in some parts of the watershed. Despite these limitations, these results can potentially be useful for managers and policy makers to identify local hotspots for land degradation to implement mitigation and adaptation measures in this watershed. Future studies could work on improving the spatial resolution and coverage of the risk assessment for providing more information on risks for land degradation. In addition, it is suggested that future research should estimate the role of other climatic factors such as humidity, and air temperature on soil erosion and dust source susceptibility. Prediction of NDVI and rainfall as the most effective factors on soil erosion and dust sources and estimated of their impacts on future water induced-soil erosion and dust sources susceptibility is also suggested for the other studies. It requires more measurements for soil erosion by water and winds to train the machine learning models.

---

## Author Comment (AC3)

**Mehdi Boroghani, Assistant Professor, Dr.**
*(Watershed Management Engineering)*
Research Center for Geoscience and Social Studies
Hakim Sabzevari University
Sabzevar, IRAN
E-mail: m.broghani@hsu.ac.ir

**Dear Prof. Dr. Lorena Grabowski**

**Editor-in-Chief**
**SOIL**

Greetings;

Attached please find the revised version of our manuscript entitled "**Mapping land degradation risk due to wind and water erosion**". Full revision has been carried out by responding to the comments and considering suggestions made by the reviewers. The implementation of those valuable comments and suggestions has significantly improved the quality of the paper for which we are grateful to the editor and the reviewers. I am also attaching a note in which major changes carried out in the paper have been explained. All revised parts have been highlighted in blue color.

I hope the emendations caused to consent the respected editor and make my paper well qualified for further processing leading to final acceptance and publication. The acknowledge receipt of the same and informing me about the final status of the paper is appreciated.
Sincerely,

Dr. M. Boroghani
Enclosure

**Reviewer 3 # Revision comments:**

The study is about "Mapping land degradation risk due to wind and water erosion". Principles and methodology are well supported. However, I would like to consider some specific comments below as minor revision. In general, there are some grammar errors throughout the manuscript. Please check the spellchecking in addition to these minor issues.

Page 5: Only 23 dusty days are used ….. out of 500 dust days, I speculate…every year: 28 dusty days (2000-2020). I would like to see a histogram with MODIS AOD (500nm, 2000- 2020) at least for one key dust station. I think all days with AOD>0.5 (500nm) are dusty days. Is the use of 23 days really a representative approach?

**Ans.** this is true that all days when the AOD is less than 0.5 as the day of dust, and also the number of days that the synoptic stations in the area have recorded as the day of dust is much higher, but this dust may be in hours of the day or It happened last night that we could not see them on MODIS images. As you know, MODIS takes pictures daily, in the morning and the afternoon at 10.30 in the morning and 2.30 in the afternoon, so the dust that occurs in the area must be at the same time as the image that we can see on MODIS image. For example, if the dust occurs at 4 pm or night, it cannot be recorded, it may also be cloudy at the same time as the dust occurrence and the dust is not visible. As a result, a large number of dust events that occurred in the area cannot be recorded on MODIS and we cannot use it, otherwise, your opinion is completely correct and the number under investigation is less than the actual number of dust that occurred in the area.

Why not showing a correlation with wind speed vs AOD for several selected observational sites? Your argumentation is not very much convincing without some demonstrations (visualizations). One could get the impression, the work is based, at the end, on a few 'beauty data' only, but does not allow solid conclusions and does not provide insight into the complex problematic.

**Ans.** We used the horizontal visibility data (dust day) of synoptic stations to identify the dust source area. Then, the days when imaging was accompanied by dust were selected and images were used. We first identified the dust on the images using dust indicators and then we determined the dust source area with visibility. First, we used remote sensing techniques and then observations. visibility data (dust day) at the synoptic station indicates the day the dust occurred in the area and no longer needs wind speed data to prove it, although correlation can also emphasize this.

Did you consider these days with visibility less than 2 km as non fog/haze days? How did you differentiate between fog and dust?

**Ans.** We only used less than 2 kilometers horizontally for dust days, not fog days. To separate these two data, we used dust meteorological codes including codes 06, 07, 08, 09 and 30 to 35.

Once per day cannot be considered as high resolution?

**Ans.** True, once-a-day imaging has a lower resolution, such as Landsat and ... but for dust studies, images need to be taken daily to capture the dust. And since the dust is occurring on a large scale and requires images to be captured on a large scale, the MODIS image is the best image for dust studies and most studies on dust. The world uses these images.

Considering that the map of dust sources for Iran was prepared by the Geological Organization, why did you use satellite images to produce it? How accurate is your generated map compared to the ground map?

**Ans.** The use of satellite images to prepare a map of dust collection areas helps us to identify all dust collection centers, especially new and smaller ones. And because we have examined a period from 2005 to 2022, as a result, centers that may have been

inactive in some years can also be identified. In the map of the dust collection centers prepared by the Geological Organization, it is in the form of a zone (the area cannot be modeled) and also they have not identified all the centers and it is general, while the map prepared by satellite images is the face is pointy and partial. Considering that Iran is a vast country, without the use of satellite images and only using field survey, the map prepared is not accurate enough, and the map of the mapping organization is not up to date and has not identified new source. Another point is that the study area is also in Iraq, as a result, the use of satellite images creates homogenous conditions and that the mapping organization was prepared only for Iran, not neighboring countries.

How to deal with the satellite data with different spatial resolutions in the study?

**Ans.** Thank you for your detailed question. In machine learning, all the layers must be pixels, and we first prepared each layer with the satellite image (MODIS, Landsat and etc.) in ENVI software, and for modeling in the ArcGIS software, we unify the pixel size layers using resampling

Line 26: The water and aeolian soil erosion maps: revised

Line 32: the risk of land degradation in an inhabited region: revised

Line 43: soil erosion in a short time: revised

Line 49: has detrimental impacts on the Earth system: revised

Line 52: therefore necessary for developing a better understanding: revised

Line 85: to the increasing dust concentration in southwest Asia: revised

Line 118: That information is extracted from data collected during an own field survey paired with a previous research (delete "a" before previous) : revised

Line 141: in the ten-year period (add hyphen for "ten-year") : revised

Line 162-163: constants taken during the initial calibration: revised

Line 168: we see dust aerosol in different colors and qualities in the MODIS images over 28 days: revised

Line 177-178: the identification and selection of appropriate dust sources and soil erosion-effective factors are necessary. : revised

Line 183: the Topographic Wetness Index (TWI), (Not Witness) : revised

Line 199: Annual rainfall (Fig. 3e) was obtained from: revised

Line 195: Mean annual rainfall was calculated using: revised

Line 261: The former is built while the RF model: revised

Line 276: three layers, namely, the input layer, the hidden layers: revised

Line 279: and the output layer is the maps of: revised

Line 304: and some indicators which were explained in section 2.1.2: revised

Line 356: similar results have been obtained in which RF with an accuracy of 45.8%: revised

Line 364: although the differences between FDA and ANN are in the statistical sense relatively small. : revised

Line 381: Distance from roads and rivers were recognized as the least important

factors: revised

Line 414: human activity is a contributing factor to the water-induced soil erosion. : revised

Line 425: the study area are at risk of soil erosion: revised

Line 428: The findings of the present study are therefore: revised

Line 431: The areas that fall under the category of both kinds ("falls" should change into "fall") : revised

Line 435: the adverse impacts of water-induced soil erosion are known: revised